# INRet: A General Framework for Accurate Retrieval of INRs for Shapes

## Abstract

Implicit neural representations (INRs) have become an important representation for encoding various data types, such as 3D objects/scenes, videos, and audio. They have proven to be particularly effective at generating 3D content, e.g., 3D scene reconstruction from 2D images, novel content creation, as well as the representation, interpolation and completion of 3D shapes. With the widespread generation of 3D data in an INR format, there is a need to support effective organization and retrieval of INRs saved in a data store. A key aspect of retrieval and clustering of INRs in a data store is the formulation of similarity between INRs that would, for example, enable retrieval of similar INRs using a query INR. In this work, we propose INRet (**INR Ret**rieve), a method for determining similarity between INRs that represent shapes, thus enabling accurate retrieval of similar shape INRs from an INR data store. INRet flexibly supports different INR architectures such as INRs with octree grids and hash grids, as well as different implicit functions including signed/unsigned distance function and occupancy field. We demonstrate that our method is more general and accurate than the existing INR retrieval method, which only supports simple MLP INRs and requires the same architecture between the query and stored INRs. Compared to 3D shape retrieval by converting INRs to other representations like point clouds or multi-view images, INRet achieves higher retrieval accuracy while avoiding the overhead of conversion.

## 1 Introduction

Implicit neural representations (INRs) have become an important approach for representing data types including images, videos, audio, and 3D content. Compared to traditional representations, INRs offer several key advantages including a compact and differential representation, and the ability to be decoded at any resolution. INRs have seen many applications including neural compression for images and videos, super-resolution and super-sampling for images and videos, and even modeling audio signals given scene geometry and listener positions (Xie et al., 2022; Su et al., 2022; Chen et al., 2020; 2022b). More importantly, INRs have emerged as a promising approach for learning and representing 3D content, including learning 3D neural radiance field (NeRF) from 2D images for novel view synthesis, combining with image diffusion models for 3D model generation, as well as representation, interpolation and completion of 3D shapes (Mildenhall et al., 2020; Poole et al., 2022; Park et al., 2019). Given the promising advantages of INRs, they are expected to become an important format for representing and storing 3D visual data. As more and more 3D visual data are generated in this format, we need a way to store, organize, and retrieve them as required.

A key aspect of being able to organize and retrieve INRs in a data store is a formulation of similarity between INRs such that in a database or data store of INRs, any stored INR can be retrieved using a query such as an image or a similar INR. For instance, accurate INR retrieval can facilitate finding a similar but more detailed model or alternative recommended models from a collection of AI-generated 3D models or reconstructed models of real scenes. Retrieval of similar INRs from a store can also be used in content creation pipelines, for scene completion, and shape enhancement. Prior work has investigated approaches to determine similarity and retrieve 3D models that are represented using traditional representations such as point clouds, meshes, and voxel grids (Qi et al., 2016; Lahav & Tal, 2020; Wang et al., 2017). These approaches typically encode the shapes into embeddings using deep neural networks, where the cosine similarity of the embeddings indicates

similarity between shapes. However, there is little research on determining similarity and enabling retrieval of INRs in a data store.

In this work, our goal is to design a method that automatically determines similarity between INRs that represent shapes in a data store. This would enable accurate retrieval of similar shapes (represented as INRs) from an INR data store using a query INR and clustering and organization of INRs representing similar shapes in the data store.

There are several challenges in enabling the accurate retrieval of shapes represented as INRs. First, INRs can have many different architectures. For example, INRs can be multilayer perceptrons (MLPs) with different activation functions or more commonly, a combination of MLPs and different types of spatial feature grids, such as octrees and hash tables (Park et al., 2019; Sitzmann et al., 2020; Takikawa et al., 2021; Müller et al., 2022). The only prior work De Luigi et al. (2023) that enables determining similarity between INRs supports an MLP-based architecture, which is not commonly used today. Thus, it is unclear how to do this for grid-based INRs and to flexibly support any architecture the query/stored INRs may have. Second, INRs can represent the same intrinsic information using different implicit functions. For example, to represent 3D shapes, signed distance function (SDF), unsigned distance function (UDF), and occupancy field (Occ) are common implicit functions that INRs can encode (Park et al., 2019; Chibane et al., 2020; Mescheder et al., 2019). Thus, it is challenging to find a general method that is effective for a wide range of INR architectures as well as any implicit function.

In this work, we investigate different approaches to enable 3D shape retrieval and similarity for INRs while addressing these challenges. We propose INRet, a framework that enables identifying similar INRs and thus accurate retrieval of similar INRs given a query INR. INRet flexibly supports INRs that use any architecture as well as any implicit function. We also compare INRet with retrieval using existing approaches for traditional representations (for example, point clouds and multi-view images) by simply converting the shape INRs into these representations for retrieval.

With INRet, we determine the similarity between INRs by first generating embeddings from each INR and then using cosine similarity between these embeddings to determine the similarity. We now describe how we generate embeddings that are general across different INR architectures and implicit functions, so they can be compared using the same cosine similarity metric.

First, we design a novel encoder that generates an embedding using the weights of the INR MLP and the learned parameters of the feature grid. This encoder supports two commonly used architectures: an octree grid (NGLOD) and a multi-resolution hash grid (iNGP) (Takikawa et al., 2021; Müller et al., 2022). The key idea behind this encoder is to encode the MLP and feature grid of the INR using an MLP encoder and a Conv3D encoder respectively. The embeddings created from these encoders are then concatenated to create the INR embedding for shape retrieval. To train these encoders, we train a decoder MLP which takes the INR embedding as input, and outputs implicit function values that approximate that of the original INR. To support INR architectures that are not octree grid-based or hash grid-based, we simply convert the INR to either of these architectures.

Second, to support different implicit functions, we use separate encoders to encode INRs with different implicit functions. They are however trained to generate embeddings that are general across different implicit functions. To do this, during the encoder training process, we generate INRs representing UDF, SDF, and Occ for *each* training 3D shape. We then apply two regularization techniques to ensure that the embeddings from the separate encoders are mapped to a common latent space. The first regularization applied is explicit L2 Loss to minimize the difference between embeddings created from INRs (with different implicit functions) representing the same shape. When training the encoders, the second regularization is to use a single common decoder that outputs a single type of implicit function value (such as UDF) for all three implicit functions. We found applying both regularizations is essential for ensuring the high retrieval accuracy of INRs across different implicit functions.

We demonstrate the effectiveness of our solution on the ShapeNet10 and Pix3D datasets. Compared to existing methods that perform retrieval of INRs directly, we demonstrate that our method enables retrieval of INRs with feature grids, which cannot be done with existing solutions. Our method achieves 10.1% higher retrieval accuracy on average than existing methods that can only retrieve shapes represented by MLP-only INRs (De Luigi et al., 2023). We show that our regularization techniques enable retrieval of INRs across different implicit functions, achieving accuracy close

to retrieval of INRs with the same implicit functions. Compared with retrieval methods applied to other representations such as point cloud and multi-view images converted from INRs, INRet achieves 12.1% higher accuracy. Except in cases where INR architecture conversion is required, INRet has lower retrieval latency as it avoids the computation overhead for the converting to other representations.

The contributions of this work are as follows. First, we pose the challenge of evaluating similarity between INRs to enable retrieval and organization of shape INRs in a data store and evaluate different techniques such as conversion to traditional formats and the creation of embeddings directly from INRs. Second, we propose a method for creating embeddings from INRs with feature grids that represent shapes that can be used for retrieval and similarity evaluation. Third, we propose regularization techniques that can generate general embeddings that be used for comparison and retrieval across INRs with different implicit functions. Fourth, we demonstrate higher retrieval accuracy for both the ShapeNet10 and Pix3D datasets compared to existing INR retrieval methods and retrieval methods by converting to other traditional representations.

## 2 BACKGROUND & RELATED WORKS

### 2.1 IMPLICIT NEURAL REPRESENTATION FOR SHAPES

Traditionally, 3D shapes have been represented with explicit representations including meshes, point clouds, and 3D voxels. Implicit Neural Representation (INR) has emerged as a novel paradigm for encapsulating shapes, employing neural networks to encode functions that implicitly represent a shape's surface. Seminal works like DeepSDF and Occupancy Networks demonstrate the feasibility of employing neural networks to encode signed distance function (SDF) or occupancy of 3D shapes (Park et al., 2019; Mescheder et al., 2019). Recent advancements extended this approach to encode unsigned distance function (UDF), showcasing higher representation quality for thinner surfaces (Atzmon & Lipman, 2020a;b; Chibane et al., 2020; Zhou et al., 2022).

**INRs with Multi-layer Perceptron.** Earlier works in INRs for shapes use simple multi-layer perceptrons (MLPs) with ReLU activations to represent the implicit functions (Park et al., 2019; Mescheder et al., 2019; Chen & Zhang, 2019; Atzmon & Lipman, 2020a;b; Chibane et al., 2020; Zhou et al., 2022). Sitzmann et al. (2020) proposed to use sinusoidal activation functions in MLPs to more efficiently encode higher frequency details. Since the implicit function is encoded in a single MLP, the MLP is usually relatively big and expensive to evaluate. The training of these MLPs to accurately represent the shapes is also time-consuming.

**INRs with Spatial Grids.** While overfitting a large MLP to a shape can be difficult and computationally expensive, recent INRs for shapes use a combination of smaller MLPs and feature grids with learnable parameters. Peng et al. introduced Convolutional Occupancy Networks, which combine a trainable 3D dense feature grid and an MLP (Peng et al., 2020). Recent works have extended this notion and applied multi-level feature grids to encode and combine information at varying levels of detail. These multi-level spatial grids can be represented as sparse octrees as seen in NGLOD, VQAD, NeuralVDB, and ROAD (Takikawa et al., 2021; 2022a; Kim et al., 2022; Zakharov et al., 2022). Müller et al. (2022) introduced the idea of using multi-level hash grids to store these features at a fixed memory budget. Both sparse octrees and hash grids have seen wide applications in implicit neural representations for shapes or for radiance fields (Tancik et al., 2023; Xie et al., 2022). Compared to INRs with only MLP, they significantly improve representation quality, as well as training and rendering speed. Our method considers INRs with or without the spatial grid for retrieval. We do so by optionally encoding the spatial grid for the INR embedding creation.

### 2.2 SHAPE RETRIEVAL

**INR Retrieval.** Numerous techniques have been developed to encode 3D shapes using INRs. The seminal work DeepSDF (Park et al., 2019) employs a shared Multi-Layer Perceptron (MLP) with varying latent codes to represent distinct shape instances. These latent codes can be used for shape retrieval, as akin shapes tend to exhibit similar codes. Nonetheless, the adoption of the shared MLP concept in subsequent research has been limited due to its compromised representation quality when contrasted with employing a dedicated MLP for each shape (Davies et al., 2020). Inr2vec De Luigi et al. (2023) introduced a method that utilizes an additional MLP to encode INR MLP weights into an

embedding, thereby enabling retrieval based on cosine similarity between embeddings from different INRs, but only applies to MLP-based INRs that are not commonly used. Compared to inr2vec, INRet not only supports INRs with spatial grids, but also enables retrieval when the query/stored INRs have different architectures or implicit functions.

**Retrieval by Converting to Traditional Representations.** Shape retrieval for traditional 3D representations has many established works with techniques proposed for voxels, meshes, point clouds and multi-view images (Wang et al., 2017; Lahav & Tal, 2020; Qi et al., 2016; Wei et al., 2020). However, these methods do not directly apply to the retrieval of INRs. A viable approach to retrieve INRs for shapes is to first transform these representations into one of the above representations, and then apply established retrieval methods. We select two representations for comparison with our method: point clouds and multi-view images because they achieve higher accuracy over retrieval with other representations (Wang et al., 2017; Lahav & Tal, 2020). In addition to higher accuracy, point-based and multi-view image-based methods also avoid the computation overhead for the voxel-based methods or the requirement for watertight surfaces for the mesh methods Wang et al. (2018); Mitchel et al. (2021). We use the state-of-the-art methods PointNeXt and View-GCN as point-based and multi-view images-based baselines for comparison.

## 3 METHODS

### 3.1 PRELIMINARY - IMPLICIT NEURAL REPRESENTATION FOR SHAPES

In this section, we introduce the different INR implicit functions and architectures we consider in this work. Consider a general distance or occupancy function $d(\cdot)$, defined for input coordinates $\boldsymbol{x} \in \mathbb{R}^3$ on the input domain of $\Omega = \{\|\boldsymbol{x}\|_\infty \leq 1 | \boldsymbol{x} \in \mathbb{R}^3\}$. The goal of INR for shape is to approximate $d(\cdot)$ by a function $f_\theta$ parameterized by a neural network.

$$f_\theta(\boldsymbol{x}) \approx d(\boldsymbol{x}), \forall \boldsymbol{x} \in \Omega. \tag{1}$$

Popular choices for the implicit function include signed distance function (SDF, $d_s(\cdot) \in \mathbb{R}$), unsigned distance function (UDF, $d_u(\cdot) \in \mathbb{R}^+$), and occupancy fields (Occ, $d_o(\cdot) \in \{-1, 1\}$) Park et al. (2019); Chibane et al. (2020); Mescheder et al. (2019). We refer to these as *implicit functions* in general for the rest of the paper. INRs are trained to minimize the difference between $f_\theta(\boldsymbol{x})$ and $d(\boldsymbol{x})$. Earlier works parameterize the function with a multi-layer perception (MLP). More recent works combine a feature grid with a smaller MLP, where the MLP takes the feature $\boldsymbol{z}$ sampled from the feature grid $\mathcal{Z}$ as input.

$$f_\theta(\boldsymbol{x}; \boldsymbol{z}(\boldsymbol{x}, \mathcal{Z})) \approx d(\boldsymbol{x}), \forall \boldsymbol{x} \in \Omega. \tag{2}$$

The feature grid $\mathcal{Z}$ has various forms, we consider sparse voxel octree (Takikawa et al., 2021) and multilevel hash grids (Müller et al., 2022) in this work. For a multi-level octree-based or hash-based feature grid, at each level $l \in \{1, ..., L\}$, the feature vector $\boldsymbol{\psi}(\boldsymbol{x}; l, \mathcal{Z})$ is interpolated (i.e., trilinear) from local features. The final feature vector $\boldsymbol{z}$ from the grid is a summation or concatenation of features from all levels. The feature vector is then optionally concatenated with $\boldsymbol{x}$ and fed to a shallow MLP to calculate the distance or occupancy value.

### 3.2 EMBEDDING CREATION FOR INR WITH FEATURE GRIDS

We determine the similarity between 3D shapes represented as INRs by converting each INR into an embedding, and the similarity between shapes is determined by the cosine similarity between these embeddings. We demonstrate our process for creating embeddings from INRs with feature grids in Figure 1. Given a trained INR with an MLP component parametrized by $\theta$ and a feature grid $\mathcal{Z}$, we use an MLP Encoder $\boldsymbol{m}$ and Conv3D Encoder $\boldsymbol{c}$ to encode the features from the MLP and grid components of the INR respectively. For the INR MLP component, the flattened weights of INR's MLP become input vectors to the MLP encoder. Because the MLP of INRs with feature grids is often small, we use all weights of the MLP as input, unlike inr2vec (De Luigi et al., 2023) which only uses the hidden layer weights. The structure of the encoder MLP is described in Appendix A.2.

For the INR feature grid, we sample $(2N)^3$ feature vectors at a fixed resolution from the feature grid, i.e. $S = \{[x_1 x_2 x_3]^T | x_i = \pm(\frac{1}{2N} + \frac{n}{N}), \forall n \in \{1, 2, \ldots, N-1\}\}$. The sampled features are used as inputs to the Conv3D encoder, a 3D convolutional network that fuses discrete spatial features with gradually increasing perception fields (see Appendix A.2 for more details). We use

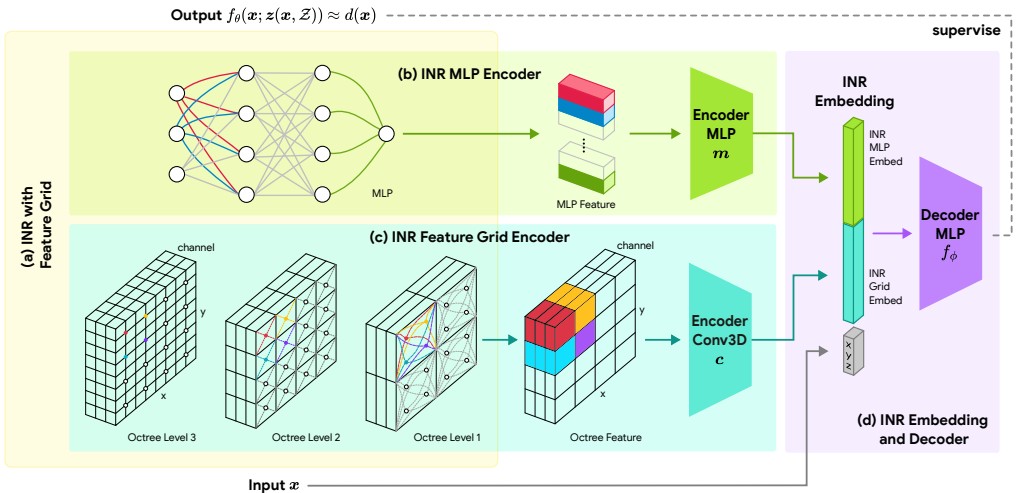

Figure 1: **Encoders trained to generate embeddings for grid-based INRs:** INRs with hash-table/octree based feature grids are used to generate embeddings for similarity calculations using encoders ($\boldsymbol{m}$ and $\boldsymbol{c}$). The encoders take the MLP weights and feature grid parameters as input to generate the embedding. The encoders are trained ahead of time using a decoder $f_\phi$ that generates the original implicit function as output.

an octree (visualized in 2D) as an example in Figure 1(**c**). Depending on the sampling resolution and resolution of the octree level, the feature is either collected directly from the corners of the voxels (Octree Level 3 in the example), or interpolated using features stored at the corners of the voxel containing the sampling location (Octree Level 2 & 1). The features collected from each level are summed together, simply adding zero if a voxel is missing (due to sparsity in the octree). The collected features are fed to a Conv3D Encoder to create the INR Grid Embedding. Similarly, for a multi-resolution hash grid based INR, we retrieve the features at the sampled points as input to the encoder. The MLP embedding and grid embedding are then concatenated to create our INR embedding.

**Training the encoders.** During the encoder training process, we feed the concatenation of the INR embedding and the input coordinate $\boldsymbol{x}$ to a decoder MLP $f_\phi$. With these two inputs to the decoder, the decoder is supervised to generate the original implicit function that represents the shape. Thus, the encoders are trained to generate embeddings that can be used to regenerate the original shape using the decoder. The following equation describes the process, where the decoder approximates the output value of the original INR:

$$f_\phi(\boldsymbol{x}; [\boldsymbol{c}(\boldsymbol{z}); \boldsymbol{m}(\theta)]) \approx f_\theta(\boldsymbol{x}; \boldsymbol{z}(\boldsymbol{x}, \mathcal{Z})), \ \forall \boldsymbol{x} \in \Omega. \tag{3}$$

**Supporting other INR architectures.** INRet assumes a separate encoder for each type of INR architecture that is supported. Our proposed encoders already support the commonly used hash table and octree-based INR architectures. A similar feature grid sampling approach can be used to also train an encoder for any new grid-based architecture. Alternatively, other architectures can still be used with the above two encoders by using a distillation technique that converts a new INR architecture into one of the two representations that we support. We describe how this can be done in Section A.3.

### 3.3 COMMON LATENT SPACE FOR INRS WITH DIFFERENT IMPLICIT FUNCTIONS

Besides different architectures, INRs can encode different implicit functions for the same underlying shape. To support multiple implicit functions of the same shape, we train separate encoders for each implicit function. To ensure that the generated embeddings map to the same latent space, we apply two regularization techniques during the encoder training process.

The first regularization applied is explicit L2 loss to minimize the difference between embeddings created from INRs for different implicit functions of the same shape. The second regularization is

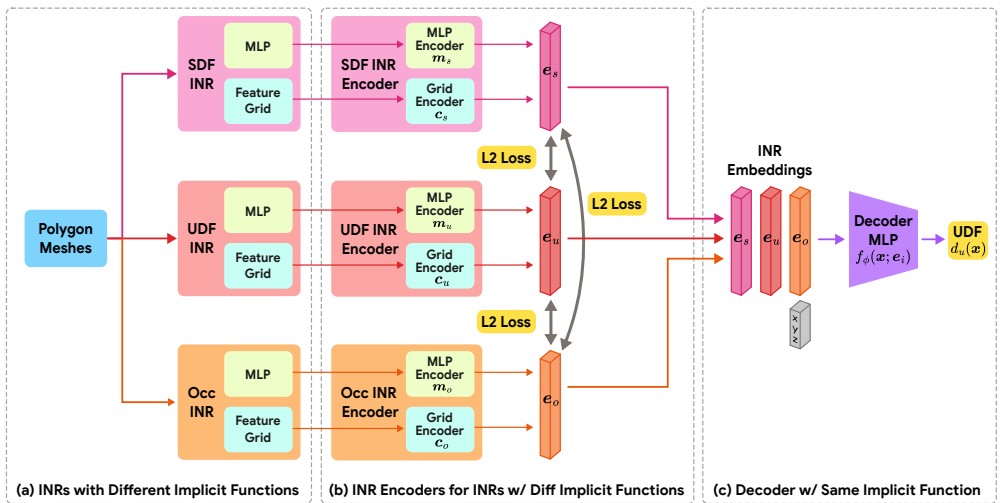

Figure 2: **INR Embedding Creation for INRs with Different Implicit Functions.** (a) For each shape, we train INRs with different implicit functions. (b) We train different encoders for INRs with different implicit functions. The differences between embeddings created by the encoders are minimized by L2 loss. (c) We feed the embeddings into a common decoder INR MLP to recreate the UDF of the original shape.

to use a single common decoder that outputs a single type of implicit function value (such as UDF) for all three implicit functions.

The overall loss function for this process is

$$\mathcal{L} = \sum_{i \in \{s,u,o\}} |f_\phi(\boldsymbol{x}; \boldsymbol{e}_i) - d_u(\boldsymbol{x})| + \lambda \sum_{i,j \in \{s,u,o\}} \|\boldsymbol{e}_i - \boldsymbol{e}_j\|^2 \tag{4}$$

where the first $s, u, o$ stands for signed/unsigned distance function and occupancy field respectively. $\boldsymbol{e}_i = [\boldsymbol{c}_i(\boldsymbol{z}_i); \boldsymbol{m}_i(\theta_i)]$ is the INR embedding for the implicit function $i$. The first part of the loss is the difference between the decoder's output with the groundtruth unsigned distance $d_u$, and the second part is the L2 loss between the INR embeddings. $\lambda$ is a hyperparameter balancing the contribution of the two parts, we found a $\lambda$ of 1 works well in practice. During the encoder training process, we create INRs for all implicit functions of each training shape to train the encoders to generate embeddings that share the same latent space.

## 3.4 CATEGORY-CHAMFER METRIC

Shape retrieval performance is traditionally evaluated based on whether the retrieved shape has the same category as the query shape (Savva et al., 2016). While this metric can evaluate the quality of retrieval based on overall shape semantics, it largely ignores similarities or differences between individual shape instances. To mitigate the shortcomings of existing metrics, we propose the Category-Chamfer metric, which evaluates whether the retrieved shape shares the same category as the query shape, and also has the least Chamfer Distance with respect to the query shape for all shapes in the category.

We choose Chamfer Distance as it can measure the distance between almost all 3D representations. Chamfer Distance is a metric that calculates similarity between two point clouds. Unlike certain metrics such as generalized Intersection over Union (gIoU) that require watertight surfaces with a clear definition of inside and outside, Chamfer Distance only requires a point cloud which can be easily converted from other 3D representations including meshes, voxels, and INRs.

The accuracy $A_C$ based on category information only is

$$A_C = \frac{\sum_{q \in Q} \delta(C(q), C(R(q)))}{|Q|} \tag{5}$$

where $Q$ is the query set, $C$ and $R$ denote the category and retrieval function respectively, the Kronecker delta $\delta(\cdot, \cdot)$ evaluates to 1 if $C(q)$ and $C(R(q))$ are the same and 0 otherwise. The accuracy is normalized by the total length $|Q|$ of the query set.

The accuracy $A_{CC}$ based on category and Chamfer Distance is

$$A_{CC} = \frac{\sum_{q \in Q}[\delta(C(q), C(R(q))) \times \delta(\text{argmin}_{s \in S} d_{CD}(q, s), C(R(q)))]}{|Q|} \tag{6}$$

where $d_{CD}$ denotes the Chamfer Distance, $S$ denotes the set of all candidates for retrieval.

Category-Chamfer is a more challenging metric compared to category-only metric, in our experiments, we find that we can leverage the Chamfer Distance between the the INR instances to achieve a high accuracy for this metric. We provide more details in Appendix E.

## 4  RETRIEVAL BY CONVERTING TO EXPLICIT REPRESENTATIONS

An alternative approach to evaluate similarity and enable retrieval of similar INRs is to first convert to an explicit representation, such as point clouds or multi-view images. This approach would enable the use of prior research to evaluate similarity between shapes represented in these traditional formats. In this work, we also evaluate the effectiveness of this approach in comparison to directly using INR embeddings. Conversion to point clouds and multi-view images from SDF INRs can be done through spherical tracing (Hart, 1996). For point cloud sampling, we start spherical tracing from randomly selected locations and directions until enough points on the surface of the object are collected (Takikawa et al., 2021). The multi-view images are also collected via spherical tracing starting from camera centers at fixed positions. For UDF INRs, we use the damped spherical tracing presented in prior work (Chibane et al., 2020) that avoids overshooting. For the occupancy values, spherical tracing is not possible so we follow the method presented in Occupancy Networks (Mescheder et al., 2019). Using occupancy values sampled at fixed resolutions from the trained INR, we combine isosurface extraction and the marching cubes algorithm to create the surface mesh of the object (Lorensen & Cline, 1987). We then perform point cloud sampling and multi-view image rendering from the constructed mesh. To generate embeddings for similarity evaluations from these formats, we use PointNeXt (Qian et al., 2022) for extracted point clouds, and View-GCN (Wei et al., 2020) for multi-view images (details are in Appendix A.4).

## 5  EVALUATION

### 5.1  EXPERIMENTAL SETUP

**Datasets.** We use ShapeNet (Chang et al., 2015) and Pix3D (Sun et al., 2018) to demonstrate the generality and robustness of our solution. For the ShapeNet10 dataset, each category has 50 models for training and 50 models for testing. For Pix3D, we use 70% of the shapes from each category as training data and 30% as testing data.

**Metrics.** We evaluate the effectiveness of our framework in identifying similar shapes in the data store by using a test INR shape to retrieve the most similar $k$ INR shapes. Following the approach in De Luigi et al. (2023), we report the mean Average Precision (mAP) as the average accuracy of retrieving a shape from the same category as the query shape across all shapes in the test set. We also report precision, recall, and F1 score as defined in the ShapeNet retrieval challenge (Savva et al., 2016) in the Appendix B.

**Baselines.** We compare against inr2vec for retrieval from INRs by directly encoding the INR weights. We also compare with PointNeXt and View-GCN by converting the trained INR to point-cloud and multi-view images, respectively.

### 5.2  INR RETRIEVAL WITH FEATURE GRIDS

To create a baseline shape INR data store, we train NGLOD (Takikawa et al., 2021) and iNGP (Müller et al., 2022) INRs with SDF to encode shapes from ShapeNet10 and Pix3D datasets.

| Method | Ours | | inr2vec | PointNeXt | View-GCN |
|---|---|---|---|---|---|
| Input Type | NGLOD | iNGP | MLP INR | Point Cloud | Multi-View Images |
| mAP @ 1 | 82.6/74.3 | **84.2/78.0** | 73.4/71.4 | 71.2/69.3 | 73.6/70.5 |
| Ret. Speed(s) | 0.034 | 0.14 | 0.062 | 0.98 | 3.05 |

Table 1: Shape Retrieval Accuracy and Speed on SDF INRs (ShapeNet10/Pix3D)

| Method | Ours | | PointNeXt | View-GCN |
|---|---|---|---|---|
| Input Type | NGLOD | iNGP | Point Cloud | Multi-View Images |
| mAP @ 1 | 76.2/71.3 | **79.2/75.5** | 70.2/67.1 | 71.4/68.2 |
| Ret. Speed(s) | 30.2 | 29.6 | 1.26 | 4.57 |

Table 2: Shape Retrieval Accuracy with 3-layer MLP-only INR as Query (ShapeNet10/Pix3D)

The encoders are trained on our split training set and used to generate embeddings for the test set. Table 1 presents mAP@1 and retrieval speed (in seconds), and additional metrics are available in Appendix B.1. Our comparison includes INRet against inr2vec, which performs retrieval on MLP-only INRs. Additionally, we compare with PointNeXt and View-GCN by converting the trained iNGP INR to point clouds and multi-view images for retrieval.

As seen in Table 1, INRet achieves the highest accuracy: on average 12.0%, 15.4%, and 12.6% higher accuracy than inr2vec, PointNeXt and View-GCN methods respectively (for iNGP INRs). Retrieving from NGLOD INRs using INRet has 3.4% lower accuracy than retrieving from iNGP INRs but still outperforms the other retrieval methods significantly. In terms of retrieval speed, INRet on NGLOD is the fastest, followed by inr2vec, which is slightly slower due to the large number of weights in the INR MLP. Compared to NGLOD, embedding generation with iNGP is slower due to the higher overhead of the hash operations during sampling. Converting to point clouds or multi-view images for retrieval with PointNeXt or View-GCN is 1-2 orders of magnitude slower than directly encoding the INR weights.

In summary, INRet enables high-accuracy retrieval of similar shape INRs. Converting to images or point clouds leads to lower accuracy due to information loss during the conversion process and incurs the latency overhead for format conversion.

## 5.3 INR Retrieval with Different Architectures

In this section, we evaluate INRet's effectiveness in retrieving shapes across INRs with an MLP-only architecture (i.e. no octree or hash table). We consider a 3-layer MLP (similar to that used by inr2vec). We apply the INR distillation technique discussed in Section 3.2 to convert the MLPs into NGLOD and iNGP so it can be used with INRet.

As depicted in Table 2, following INR distillation, we achieve an average accuracy of 73.8% and 77.4% respectively for NGLOD and iNGP encoders, surpassing inr2vec by 6.3% and 14.1% on the respective datasets. Our method also performs better than converting to point cloud or multi-view images. This highlights the robustness of our approach in adapting to different architectures not directly supported by the encoder. While format conversion introduces some overhead (approximately 30 seconds), a potential speed-accuracy tradeoff could be explored by converting to point clouds/images when INRet lacks a pre-trained encoder for a new architecture.

## 5.4 INR Retrieval with Different Implicit Functions

In this section, we evaluate the effectiveness of our method in performing INR retrieval across INRs with different implicit functions (i.e., UDF, SDF and Occ). We compare against inr2vec and point-based and image-based methods, using PointNeXt and View-GCN.

As seen in Table 3, using our method to retrieve iNGP INRs with different implicit functions achieves the highest 82.0% accuracy, which is higher than the accuracy achieved with inr2vec, PointNeXt, and View-GCN. In particular, inr2vec achieves very low accuracy (around 10%) for retrieving INRs with different implicit functions. As seen in Table 4, using INRet to retrieve iNGP

| | | Retrieval INR | | |
|---|---|---|---|---|
| | | UDF | SDF | Occ |
| Query | UDF | 80.2/83.0/68.8/72.0/70.8 | 81.4/79.0/10.4/61.8/72.6 | 78.8/80.4/ 8.8/58.2/68.4 |
| | SDF | 82.2/81.8/11.4/62.2/70.2 | 83.4/84.6/70.2/67.2/69.4 | 79.2/82.4/10.4/56.2/68.8 |
| | Occ | 76.0/81.0/ 9.2/55.4/62.6 | 77.0/82.6/10.4/56.2/61.8 | 76.8/83.0/69.4/51.2/66.4 |
| Average | | 79.4/**82.0**/29.9/60.0/67.9 | | |

Table 3: Shape Retrieval Accuracy on Different Implicit Functions INRs for ShapeNet10 (NGLOD / iNGP/ inr2vec / PointNeXt / View-GCN)

| | | Retrieval INR | | |
|---|---|---|---|---|
| | | UDF | SDF | Occ |
| Query | UDF | 83.4/83.4/83.0 | 9.4/52.4/79.0 | 10.8/51.8/80.4 |
| | SDF | 10.8/57.8/81.8 | 82.4/81.4/84.6 | 9.6/53.2/82.4 |
| | Occ | 11.4/65.4/81.0 | 10.2/53.2/82.6 | 81.6/82.4/83.0 |
| Average | | 34.0/64.6/**82.0** | | |

Table 4: Shape Retrieval Accuracy on iNGP INRs for ShapeNet10 (**No Regularization / + L2 Regularization / + L2 Regularization** & **Common Decoder**)

with different implicit functions also achieves very low accuracy (around 10%) if no regularization is used. The average accuracy for retrieval improves significantly if the L2 regularization (64.6% accuracy) and both regularizations (82.0% accuracy) are applied. We provide results on the Pix3D dataset, and speed for the conversion for the different implicit functions in Appendix B.2.

## 6 CONCLUSION

In this work, we presented a new framework for determining similarity between INRs that can be used for accurate retrieval of INRs from a data store. We proposed a new encoding method for INRs with feature grids including the octree and hash table based grids. By using L2 loss and a common decoder as regularizations, INRet also enables the retrieval of INRs across different implicit functions. On ShapeNet10 and Pix3D datasets, INRet demonstrates more than 10% improvement in retrieval accuracy compared to prior work on INR retrieval and retrieval by conversion to point cloud and multi-view images. Compared to point cloud and multi-view image retrieval methods, INRet is also faster by avoiding the conversion overhead when retrieving INRs with same or different implicit functions.

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

# A  ARCHITECTURE AND TRAINING DETAILS

## A.1  INR ARCHITECTURE AND TRAINING DETAIL

**INR Training Losses.** We apply different loss functions for different implicit functions. For the signed distance function, we follow the method in Takikawa et al. (2021).

$$\mathcal{L}_s(f_\theta(\boldsymbol{x}), d_s(\boldsymbol{x})) = \|f_\theta(\boldsymbol{x}) - d_s(\boldsymbol{x})\|^2 \tag{7}$$

For the unsigned distance function, we follow the method in Chibane et al. (2020)

$$\mathcal{L}_u(f_\theta(\boldsymbol{x}), d_u(\boldsymbol{x})) = |f_\theta(\boldsymbol{x}) - d_u(\boldsymbol{x})| \tag{8}$$

For the occupancy field, we also apply an L1 loss similar to the unsigned distance function.

**INR Architectures.** We implement the INR architectures in this work using the NVIDIA Kaolin Wisp library  (Takikawa et al., 2022b). We follow the default configurations for the NGLOD and iNGP architectures. For NGLOD, we use an octree with 6 levels, but do not store features in the first 2 levels following the default configuration. We use a feature size of 8 for each level. For iNGP, we utilize 4 levels for the hash grid. The minimum and maximum grid resolutions are set to 16 and 512, respectively, with a maximum hashtable size of $2^{19}$ for each level. The feature size is 2 for each level. For both grids, we initialize the grid features by sampling from a normal distribution with a mean of 0 and a standard deviation of 0.01. For the MLP-only INR, we follow the configuration in De Luigi et al. (2023), and train a SIREN INR with 4 hidden layers and 512 hidden nodes Sitzmann et al. (2020). The MLP uses sine activation functions.

**INR Training.** We use the polygon meshes from the ShapeNet10 and Pix3D datasets to generate SDF, UDF and Occ values to train the INRs. We use the sampling method from Kaolin Wisp. We sample $5 \times 10^5$ points for each shape per epoch of training. We sample $10^5$ points uniformly in the domain $\Omega = \{\|\boldsymbol{x}\|_\infty \leq 1 | \boldsymbol{x} \in \mathbb{R}^3\}$, $2 \times 10^5$ on the surface of the shape, and $2 \times 10^5$ near the surface using normal distribution with a variance of 0.015. We use the same input coordinates for the training of all INRs. We train the grid-based INRs for 10 epochs and the MLP-only INRs for 100 epochs. We use Adam optimizer with a learning rate of $1e-3$  (Kingma & Ba, 2014).

**INR Initialization.** For the MLP component of INR, we follow inr2vec's method and initialize the MLP with the same weights, which was proven to be essential for the embedding from the MLP component to be meaningful for shape retrieval (De Luigi et al., 2023). However, we observe no such restriction on the initialization of the feature grid since the feature grids are initialized with very small random values to begin with.

## A.2  INR ENCODER DETAILS

**INR MLP Encoder.** We use an MLP encoder to convert the weights in the INR MLP to an embedding. To encode MLP-only INRs, we use MLP encoder with architecture identical to inr2vec. The MLP encoder consists of 4 linear layers, each followed by batch normalization and ReLU activation  (Ioffe & Szegedy, 2015). The final layer is a max pooling layer that produces an embedding of length 1024, we refer readers to  De Luigi et al. (2023) for more details. To encode the MLP component of a grid-based INR, we reduce the hidden size of all layers by half, using hidden layers with hidden size of 256, 256, 512 and 512. This creates an INR MLP embedding with length 512.

**INR Conv3D Encoder.** The Conv3D Encoder consists of five 3D convolution operations. Each convolution has a kernel size of $(2, 2, 2)$ and a stride of $(2, 2, 2)$ to gradually reduce the spatial resolution. Each convolution doubles the channel size and is followed by a group normalization and ReLU activation  (Wu & He, 2018). The last layer is a linear layer mapping output from the convolution to an INR Grid Embedding with a length of 512. Combined with the INR MLP Embedding, the total INR Embedding length for the grid-based INR is 1024, identical to the embedding length for the MLP-only INR in inr2vec.

**INR Encoder Training.** We use the same input sampling process as in the INR training process. We follow the procedure in De Luigi et al. (2023), using the Adamw optimizer with $1e-4$ learning rate and $1e-2$ weight decay (Loshchilov & Hutter, 2017). Note that our decoder MLP $f_\phi$ shares the same architecture with De Luigi et al. (2023). Only the encoders are needed for generating the INR embeddings. The decoder is only used during the training of the encoders.

Figure 3: **INR Embedding Creation for INRs with Different Architectures**

## A.3 INR DISTILLATION

Changes such as feature size dimension for the feature grid and number of hidden nodes in the MLP can cause dimension mismatches, making the encoders trained for specific architectures unusable. However, since one may want to use different architecture configurations for speed and representation quality trade-offs, or use an architecture not yet invented in the future, we need to create a general solution that does not have strict requirements on the architecture of INR.

To this end, we rely on the property that INRs for shapes are designed to output a distance or occupancy value given an input coordinate. For a given source INR with unknown architecture, to create an embedding from it for retrieval, we use the source INR $f_s$ as an oracle to create input coordinate and output value pairs to train an INR $f_\theta$ with the architecture our encoders can understand. We call this the INR distillation technique and visualize it in Figure 3.

$$f_\theta(\boldsymbol{x}; \boldsymbol{z}(\boldsymbol{x}, \mathcal{Z})) \approx f_s(\boldsymbol{x}), \ \forall \boldsymbol{x} \in \Omega. \tag{9}$$

In general, there is no limitation on either the source or target INR, or the type of the output value (distance or occupancy). In Section 5.3, we demonstrate that for a source MLP-only INR, distilled to an INR with feature grid can actually improve the retrieval accuracy compared to performing retrieval with the embeddings created from the MLP-only INR.

## A.4 EXPLICIT REPRESENTATION TRAINING AND ENCODING

**PointNeXt.** For training of the PointNeXt (Qian et al., 2022), we use the point clouds containing 2048 points sampled from the surface of the INRs representing shapes in the training set. We follow the training procedure in PointNeXt and use the PointNeXt-S variant. The training is supervised using the class of the shape. After training, we remove the classification head, and use the output from the PointNeXt background to create embeddings with length 512.

**View-GCN.** We use images rendered from the INRs representing the training shapes to train the View-GCN (Wei et al., 2020). We render 12 images at a resolution of 224 x 224. The images are rendered from virtual cameras placed 3 units of distance away from the object center with a 0.65 elevation on a circular trajectory. We follow the training procedure in View-GCN. The training is also supervised using the class of the shape. We remove the classification head after training, and use the output with length 1536 as the embedding for shape retrieval.

## B ADDITIONAL RESULTS

## B.1 INR RETRIEVAL WITH FEATURE GRIDS

In this section, we provide additional results for the experiment in Section 5.2. We show the mean Average Precision (mAP@k) at different numbers of k following the method in (De Luigi et al., 2023). We also report the precision, recall, and F1 score following the definition in ShapeNet (Savva et al., 2016). Note that for the ShapeNet10 dataset, since the number of models in each category is the same, the precision, recall and F1 score are the same, and thus we only report the F1 score. Our method achieves higher scores for almost all metrics across both ShapeNet10 and Pix3D datasets over inr2vec, PointNeXt and View-GCN. This demonstrates that our method is not only able to correctly retrieve the most similar shape, but also retrieves more shapes that belongs to the same category as the query shape as seen by the higher F1 score.

| Method | Ours | | inr2vec | PointNeXt | View-GCN |
|---|---|---|---|---|---|
| Input Type | NGLOD | iNGP | MLP INR | Point Cloud | Multi-View Image |
| mAP @ 1 | 82.6 | **84.2** | 73.4 | 68.0 | 71.6 |
| mAP @ 5 | **94.4** | **94.4** | 89.8 | 87.2 | 88.2 |
| mAP @ 10 | 96.4 | **96.6** | 92.4 | 89.6 | 90.4 |
| F1 @ 10 | 80.8 | **81.8** | 72.0 | 67.8 | 70.4 |

Table 5: Shape Retrieval Accuracy Metrics on ShapeNet10

| Method | Ours | | inr2vec | PointNeXt | View-GCN |
|---|---|---|---|---|---|
| Input Type | NGLOD | iNGP | MLP INR | Point Cloud | Multi-View Image |
| mAP @ 1 | 76.5 | **78.0** | 71.4 | 66.3 | 68.5 |
| mAP @ 5 | 88.9 | **93.8** | 91.0 | 81.5 | 87.2 |
| mAP @ 10 | 92.6 | 94.3 | **95.5** | 88.4 | 93.3 |
| P @ 10 | 66.7 | **68.0** | 62.3 | 59.9 | 61.0 |
| R @ 10 | 75.2 | **76.1** | 69.0 | 68.5 | 70.3 |
| F1 @ 10 | 70.7 | **71.9** | 65.3 | 63.9 | 65.2 |

Table 6: Shape Retrieval Accuracy Metrics on Pix3d

## B.2  INRs with Different Implicit Functions

| | | Retrieval INR | | |
|---|---|---|---|---|
| | | UDF | SDF | Occ |
| Query | UDF | 69.4/70.4/66.7/61.2/68.5 | 71.6/72.8/12.2/59.9/66.4 | 71.6/71.6/10.7/60.8/61.2 |
| | SDF | 67.9/72.8/12.4/61.4/67.4 | 74.1/79.0/71.5/62.3/67.2 | 67.9/69.1/11.9/54.4/59.7 |
| | Occ | 69.1/67.9/13.1/57.6/60.4 | 72.3/74.1/11.8/56.8/60.9 | 68.9/69.1/65.4/58.2/61.3 |
| Average | | | 70.3/**71.9**/30.6/59.2/63.7 | |

Table 7: Shape Retrieval Accuracy for Different Implicit Functions INRs on Pix3D (**NGLOD** / **iNGP**/ **inr2vec** / **PointNeXt** / **View-GCN**)

We show additional results for INR retrieval across different implicit functions on the Pix3D dataset in Table 7. Similar to the results on the ShapeNet10 dataset, our method demonstrates higher accuracy for retrieval across INRs with different implicit functions than inr2vec, PointNeXt and View-GCN.

| | | Retrieval INR | | |
|---|---|---|---|---|
| | | UDF | SDF | Occ |
| Query | UDF | 80.2/83.0 | 91.2(+  9.8)/88.4(+  9.4) | 86.6(+  7.8)/87.4(+  7.0) |
| | SDF | 92.0(+  9.8)/93.2(+11.4) | 83.4/84.6 | 90.2(+11.0)/92.8(+10.4) |
| | Occ | 85.6(+  9.6)/89.4(+  8.4) | 87.8(+10.8)/92.6(+10.0) | 76.8/83.0 |

Table 8: Shape Retrieval Accuracy for Different Implicit Functions INRs on ShapeNet10, Allowing Retrieval of Same Shape NGLOD(+Improvement)/iNGP(+Improvement)

Normally, we exclude the INR representing the same shape from being retrieved when measuring the mAP, otherwise, the query embedding always have the highest cosine similarity with itself. In Table 8, we show the accuracy of INR retrieval across different implicit functions by allowing retrieval of INR (with a different implicit function) representing the same shape. As seen in the table, there is around 10% improvement in retrieval accuracy. This shows that in many cases, the retrieved shape is the same shape as the query shape, but just represented with a different implicit function.

|  | UDF | SDF | Occ |
|---|---|---|---|
| Point Cloud | 1.26 | 0.98 | 1.82 |
| Multi-View Images | 4.13 | 3.05 | 3.67 |

Table 9: Conversion Speed (seconds) from INR with Different Implicit Functions to Different Representations

In Table 9, we show the conversion speed of converting different representations to point clouds and multi-view images. As required by View-GCN, 12 images are rendered, and as the result it is more expensive than sampling a single point cloud. Conversion to point cloud or images is also more expensive for UDF compared to SDF due to the use of damped spherical tracing.

## B.3 OTHER INR ARCHITECTURES

Despite the wide application of NGLOD and iNGP architectures due to their speed and reconstruction quality, other INR architectures also exist. For example, the triplane feature grid, which uses a combination of 2D matrices to represent features in a 3D volume, has been adapted in works such as TensorRF (Chen et al., 2022a) and EG3D (Chan et al., 2021) to represent neural radiance fields and 3D shapes. MLP-only architectures, despite being slower to train and having lower reconstruction quality, may be used under certain circumstances in the future. We apply our method to both MLP-only and triplane architectures. We use the MLP-only INR architecture configuration in De Luigi et al. (2023). For the triplane, we follow the default configuration from the open-source Kaolin Wisp library Takikawa et al. (2022b). We train SDF, UDF, and Occ INRs and follow the experiment setup described in Section 5.4.

|  |  | Retrieval INR | | |
|---|---|---|---|---|
|  |  | UDF | SDF | Occ |
| Query | UDF | 83.0/68.6/80.8 | 79.0/66.2/79.4 | 80.4/66.6/79.2 |
|  | SDF | 81.8/67.8/81.2 | 84.6/70.0/82.4 | 82.4/67.4/79.6 |
|  | Occ | 81.0/67.2/79.8 | 82.6/68.0/79.4 | 83.0/78.6/80.0 |
| Average | | **82.0**/67.8/80.2 | | |

Table 10: Shape Retrieval Accuracy for Different Implicit Function INRs on ShapeNet10 (iNGP/MLP-only/Triplane)

From Table 10, we observe that our method enables retrieval for all INR architectures, including the MLP-only architecture. Without applying our method, the average retrieval accuracy for MLP-only INR is only 29.9% as seen in Table 3. Using the regularizations of INRet, we achieve a 2.27X retrieval accuracy of 67.8%. Retrieval of INR with triplane feature grids achieves an accuracy of 80.2%, similar to the performance of retrieval with NGLOD and iNGP INRs. This demonstrates that our method is robust to the specific architecture of the feature grid.

## B.4 CHOICE OF COMMON DECODER IMPLICIT FUNCTION

In this section, we evaluate the performance of INRet with different common decoders. In Section 5.4, we used the UDF decoder as the common decoder, which takes the INR embedding as input (regardless of the original INR function's implicit function). In Table 11, we show the retrieval accuracy when the common decoder is required to output different implicit functions during training.

| | | Retrieval INR | | |
|---|---|---|---|---|
| | | UDF | SDF | Occ |
| Query | UDF | **83.0**/80.8/79.4 | 79.0/81.2/78.8 | 80.4/79.8/80.4 |
| | SDF | 81.8/81.2/80.0 | 84.6/**85.8**/83.6 | 82.4/82.4/82.6 |
| | Occ | 81.0/79.6/80.8 | 82.6/**82.8**/81.4 | 83.0/83.2/**83.4** |
| Average | | **82.0**/81.9/81.2 | | |

Table 11: Shape Retrieval Accuracy for iNGP INRs on ShapeNet10 with Different Common Decoder Choices (UDF/SDF/Occ)

As seen in Table 11, the average retrieval accuracy for different choices of common decoders is fairly close. The UDF common decoder had the highest accuracy of 82.0% while the lowest, the Occ common decoder, is only 0.8% behind in accuracy. However, we observe that if the INR's implicit function is the same as the common decoder's output, the retrieval accuracy tends to be higher. For example, for SDF to SDF retrieval, the highest retrieval accuracy of 85.8% is achieved when the common decoder's output is also an SDF. The trend also applies to UDF to UDF retrieval and Occ to Occ retrieval. In addition, for retrieval across INRs with different implicit functions, the retrieval accuracy tends to be higher if the query or retrieval INR's implicit function is the same as the common decoder's output type. Despite these tendencies, our method is generally relatively robust to the choice of the common decoder's output.

## B.5  EXPLICIT REGULARIZATION WEIGHT

In this section, we evaluate the performance when different weights are applied to the explicit L2 regularization. In INRet, the explicit L2 regularization is simultaneously applied to 3 different pairs: UDF-SDF, UDF-Occ and SDF-Occ. In the main results presented in the paper, the weighting is the same for all pairs. In Table 12, we show the retrieval accuracy when the weights are different. For example, the *211* weight means the UDF-SDF loss is multiplied by 2 before being added to the total loss, while the UDF-Occ and SDF-Occ are multiplied by 1.

| | | Retrieval INR | | |
|---|---|---|---|---|
| | | UDF | SDF | Occ |
| Query | UDF | 83.0/82.6/82.0/82.8 | 79.0/80.2/78.4/80.4 | 80.4/80.6/81.0/79.8 |
| | SDF | 81.8/81.8/82.0/81.0 | 84.6/84.0/83.8/84.8 | 82.4/81.4/81.8/80.8 |
| | Occ | 81.0/81.2/80.8/80.6 | 82.6/82.0/83.0/82.6 | 83.0/82.6/83.2/82.8 |
| Average | | **82.0**/81.8/81.8/81.7 | | |

Table 12: Shape Retrieval Accuracy for iNGP INRs on ShapeNet10 with Different Explicit L2 Regularization Weights for *UDF-SDF, UDF-Occ, SDF-Occ* (*111/211/121/112*)

From Table 12, we observe that our method is robust with respect to the specific choice of weight multipliers. For the INR encoder training, we used the Adam optimizer which has an adaptive learning rate on individual weights of the network, eliminating the need for careful fine-tuning on the weight multipliers Kingma & Ba (2014).

## B.6  L2 NORM FOR UDF COMMON DECODER

We test whether using L2 normalization for a common decoder with UDF loss type instead of L1 has an impact on the accuracy. We present the results in Table 13. From the table, we show that using L2 loss decreases the retrieval accuracy slightly compared to using the L1 loss. We note that the retrieval accuracy of individual loss function to loss function pairs can fluctuate quite significantly. For example, the Occ-SDF retrieval accuracy dropped 3.6% (from 82.6% to 79.0%). This is different from the result in Table 12 where the weighting of the explicit regularization had minimal impact on the retrieval accuracy.

|  |  | | Retrieval INR | |
|---|---|---|---|---|
|  |  | UDF | SDF | Occ |
| Query | UDF | 83.0/82.5 | 79.0/81.0 | 80.4/80.6 |
|  | SDF | 81.8/81.6 | 84.6/84.6 | 82.4/81.8 |
|  | Occ | 81.0/81.4 | 82.6/79.0 | 83.0/79.6 |
|  | Average | | **82.0**/81.3 | |

Table 13: Shape Retrieval Accuracy for iNGP INRs on ShapeNet10 with UDF Common Decoder L1/L2 Loss Choice

### B.7 SUMMATION AND CONCATENATION OF FEATURES

In the main results, the Conv3D encoder takes in the summation of features from NGLOD feature grid and the concatenation of features from iNGP feature grid. We do so because summation and concatenation of features are used in the original NGLOD INR and iNGP INR respectively.

Following Eq. 2, for NGLOD, the features from the multi-level octree feature grid are summed before fed into the MLP, i.e.

$$z(x, \mathcal{Z}) = \sum_{l}^{L} (\psi(x; l, \mathcal{Z})) \tag{10}$$

For iNGP, the features are concatenated instead, i.e.

$$z(x, \mathcal{Z}) = [\psi(x; 0, \mathcal{Z}), \psi(x; 1, \mathcal{Z}), \dots, \psi(x; L, \mathcal{Z})] \tag{11}$$

For NGLOD, features stored in different levels of the octree capture varying levels of geometry detail. Therefore, using summation allows adding finer surface information (deeper level) to the coarser overall shape (upper level) Takikawa et al. (2021). For iNGP, the features stored in the hash grid inevitably suffer from hash collision. The authors argued that using features from all levels would allow the MLP to mitigate the effect of hash collision dynamically Müller et al. (2022). Using the octree feature grid, NGLOD does not suffer from the hash collision issue.

Following the experiment setting in Section 5.2, we test the retrieval accuracy when we use features in a way different from how it was used in the original INR architecture.

| INR Arch. | NGLOD | | iNGP | |
|---|---|---|---|---|
| Feature Comb. | Sum (Original) | Concat (Modified) | Concat (Original) | Sum (Modified) |
| mAP @ 1 | **82.6** | 67.8 | **84.2** | 30.4 |

Table 14: Shape Retrieval Accuracy on ShapeNet10 when Features are Summed or Concatenated from the Feature Grids

As shown in Table 14, both methods experienced a significant drop in retrieval accuracy if the features were not used in a way consistent with the original INR. For NGLOD, the concatenation of features leads to an accuracy drop of 14.8%. We note that the concatenation of features in this case actually means more features being passed to the Conv3D encoder for INR embedding creation. However, since the finer level features were never used alone in the original NGLOD INR training, we hypothesize the Conv3D encoder may be overfitting to these finer level features that might be noisy when used standalone. For iNGP, the retrieval accuracy is dropped by 53.8% since the summation of features leads to a significant loss of information.

## C RECONSTRUCTION QUALITY AND RETRIEVAL VISUALIZATION

### C.1 RECONSTRUCTION QUALITY

In this section, we provide additional details on the quality of reconstruction of the trained INRs with respect to the original mesh. For UDF INRs, we measure the Chamfer Distance (C.D.) at 130,172 points, following the same sampling method used in Takikawa et al. (2021). However,

instead of regular spherical tracing, we apply damped spherical tracing similar to Chibane et al. (2020). For SDF and Occ INRs, we use vanilla spherical tracing without damping, and we also measure generalized Intersection over Union (gIoU) which calculates the intersection of the inside of two watertight surfaces with respect to their union. We do not measure gIoU for UDF INRs as there is no notion of inside and outside.

| INR Arch. | NGLOD | | | iNGP | | | MLP | | |
|---|---|---|---|---|---|---|---|---|---|
| Implicit Func. | SDF | UDF | Occ | SDF | UDF | Occ | SDF | UDF | Occ |
| C.D. ShapeNet | 0.0168 | 0.0122 | 0.0210 | 0.0147 | **0.0119** | 0.0223 | 0.0354 | 0.0344 | 0.0389 |
| C.D. Pix3D | 0.0183 | 0.0125 | 0.0213 | 0.0146 | **0.0120** | 0.0241 | 0.0367 | 0.0351 | 0.0392 |
| gIoU ShapeNet | 84.2 | NA | 81.4 | **86.2** | NA | 82.1 | 77.3 | NA | 75.2 |
| gIoU Pix3D | 85.5 | NA | 82.2 | **86.5** | NA | 82.3 | 77.5 | NA | 74.9 |

Table 15: Shape Reconstruction Quality of different INRs on ShapeNet and Pix3D

As seen in Table 15, both the NGLOD and iNGP achieve higher reconstruction quality than the MLP INRs. These INRs with feature grids are not only superior at representing shapes with higher fidelity but also lead to higher retrieval accuracy.

Since INR with feature grids have both higher reconstruction quality and higher retrieval accuracy, one may wonder if these are correlated. We perform another experiment, where the iNGP is only trained for only 2 epochs, leading to reconstruction quality lower than the MLP-only INR. As seen in Table 16, the retrieval accuracy for iNGP significantly drops when the INRs are undertrained. However, retrieval with iNGP @ 2 epochs still has 5.4% higher accuracy compared to retrieval with MLP-only INR. The MLP-only INR lacks the features stored spatially in the feature grid, which is very useful for improving retrieval accuracy.

| Method | Ours | | inr2vec |
|---|---|---|---|
| Input Type | iNGP | iNGP @ 2 Epoch | MLP INR |
| mAP @ 1 | **84.2** | 78.8 | 73.4 |
| C.D. | 0.0168 | 0.0371 | 0.0354 |

Table 16: Shape Retrieval Accuracy and Reconstruction Quality Comparison for Different INR Architectures (SDF) on ShapeNet10

## C.2 RETRIEVAL VISUALIZATION

In this section, we qualitatively evaluate the performance of retrieval from different methods. We show the shape retrieved using different methods, as well as across INRs with different implicit functions.

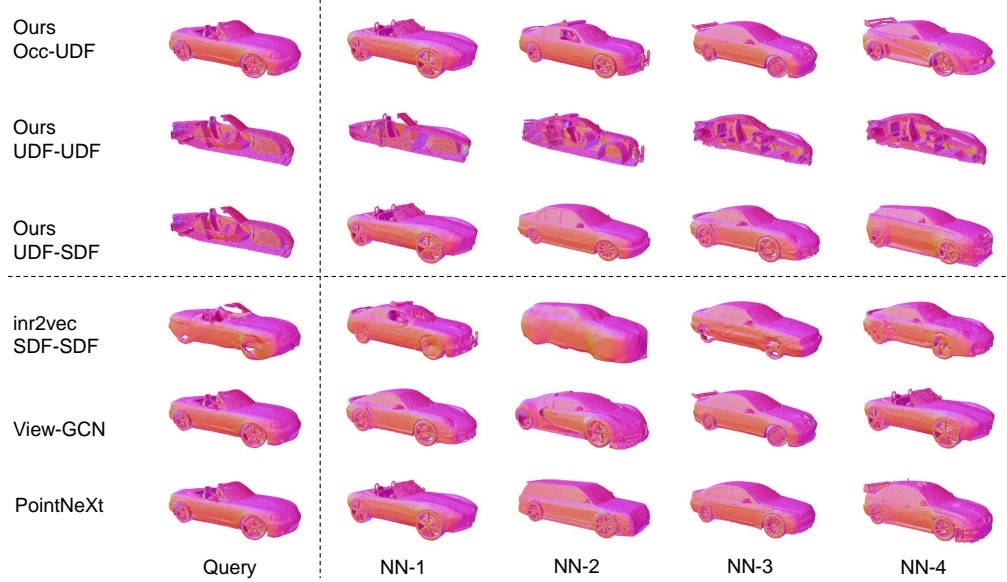

Figure 4: Retrieval Qualitative Comparison.

Figure 4 shows the query and retrieved results for the car class in ShapeNet. The car class is a relatively easy class where most models retrieve shapes from the shape category. However, as we can see from the figure, our method consistently retrieves the other convertible from the dataset as the top candidate while most other methods fail to do so. PointNeXt retrieves the convertible as the top candidate but the other models retrieved do not resemble the query shape very well.

Besides the retrieval similarity, we also demonstrate the ability to retrieve watertight surfaces (represented with SDF INR) from surfaces with multiple layers (represented with UDF INR). For the *Ours UDF-UDF* and *Ours UDF-SDF*, we show renderings of the same query car, but with the middle cut open when the shape is represented using a UDF INR. We can see that the UDF INR can capture the details inside the car, and is also able to retrieve other cars correctly.

Compared with the renderings demonstrated in the row *inr2vec SDF-SDF* which used an MLP-only INR to represent the underlying shape. Our method uses iNGP to represent the shape thus capturing more details.

# D    EMBEDDING SPACE T-SNE VISUALIZATION

In figure 5, we provide the t-SNE plot of the INR embeddings created by the INR encoders trained with and without the decoders. Without the common decoder, the embeddings for shapes belonging to the same category are much more spread out, this is likely due to the different decoders requiring the INR embedding for the same shape to be used for different purposes (decoding UDF, SDF, or Occ). This misses the regularization from the common decoder that further minimizes the difference between embeddings of the same shape represented with INRs with different implicit functions.

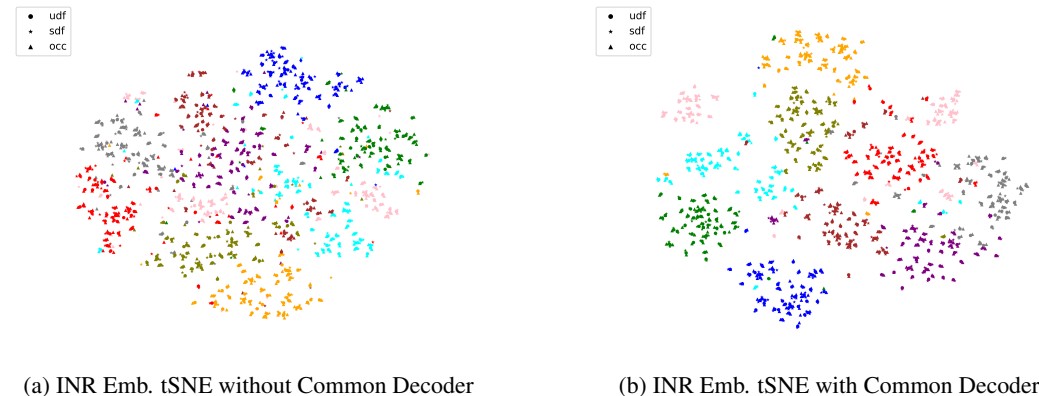

(a) INR Emb. tSNE without Common Decoder    (b) INR Emb. tSNE with Common Decoder

Figure 5: tSNE Plot for INR Embedding Without and With the Common Decoder for INR Encoder Training

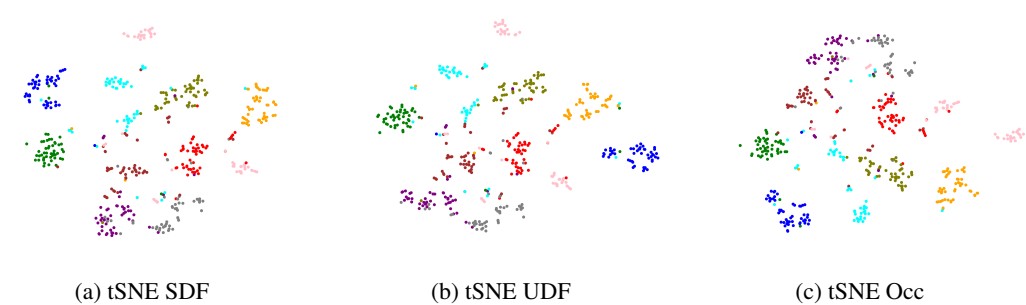

(a) tSNE SDF              (b) tSNE UDF              (c) tSNE Occ

Figure 6: tSNE Plot for INR Embedding from INRs with Different Implicit Functions. All Regularizations Applied

# E    CATEGORY-CHAMFER METRIC

## E.1    CATEGORY-CHAMFER RETRIEVAL ACCURACY BY EMBEDDING COSINE SIMILARITY

Compared with the category-only accuracy, achieving high accuracy as measured by the Category-Chamfer metric is more challenging. By simply comparing the cosine similarity between embeddings, neither INRet or existing methods such as PointNeXt perform well for this new metric. We exclude View-GCN from this evaluation since it may not require an actual 3D model to perform the retrieval and thus may not be able to calculate Chamfer Distance given its input. Following the procedure in Section 5.3, we evaluate the Category-Chamfer accuracy.

| Method | Ours | | PointNeXt |
|---|---|---|---|
| Input Type | NGLOD | iNGP | Point Cloud |
| $A_C$ | 82.6 | **84.2** | 71.2 |
| $A_{CC}$ | 21.2 | 23.2 | **28.4** |

Table 17: Retrieval Accuracy (Category, Category-Chamfer) on ShapeNet10

We calculate the ground truth Chamfer Distance at 131072 points following the same sampling method from Takikawa et al. (2021). From Table 17, we observe that the Category-Chamfer accuracy for all methods is very low. The highest accuracy is achieved by PointNext at 28.4%, far below its category-only accuracy of 71.2%. In the next section, we provide a solution for increasing the Category-Chamfer retrieval accuracy while avoiding significant runtime overhead.

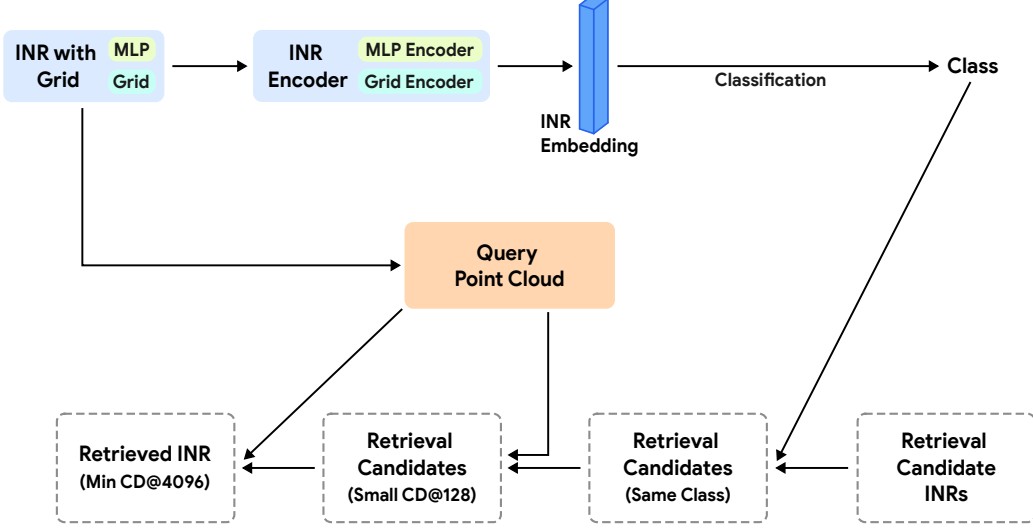

Figure 7: Hierarchical Sampling Retrieval Method

## E.2 HIERARCHICAL SAMPLING

Deep learning-based shape retrieval methods usually involve calculating an embedding for the input shape, and retrieval is done by comparing the cosine similarity between the embeddings. However, as seen in Table 17, these methods do not perform well on the Category-Chamfer metric. Unlike cosine similarity which can be easily computed in a batched manner, Chamfer Distance requires comparison between individual point clouds. A naive solution is to calculate the Chamfer Distance with all other shapes within the same category. However, such a naive method would require extensive computation, scaling linearly with the size of the dataset for retrieval.

To this end, we propose a Hierarchical Sampling approach, visualized in Figure 7. We found that the Chamfer Distance at a small number of points (128) is an effective proxy for the Chamfer Distance at a large number of points (4096). Although we calculate the groundtruth Chamfer Distance at 131072 points following typical values used for evaluation of 3D shape reconstruction quality Takikawa et al. (2021), we found that in terms of ranking of shape by Chamfer Distance, 4096 points is sufficient. For INRet, we use the frozen INR Embeddings to train an MLP for classification, following the same settings as De Luigi et al. (2023). We use E-Stitchup to augment the input with interpolations of INR embeddings from the same class Wolfe & Lundgaard (2019). For PointNeXt, we use the trained PointNeXt to do the classification.

We present the result of the retrieval in Table 18. For naive retrieval, we directly sample points and calculate the Chamfer Distance at 4096 points between the query INR and all candidate INRs. For Hierarchical Sampling retrieval, we first sample points and calculate the Chamfer Distance at 128 points between the query INR and all candidate INRs. We further calculate the Chamfer Distance at 4096 points for all INRs with a small Chamfer Distance at 128 points. We define small by the INR having Chamfer Distance within 3 times of the smallest Chamfer Distance between query INR and all candidate INRs. This is a very generous bound and ensures a 100% recall on our dataset. The accuracy is effectively only limited by the classification accuracy.

| Method | Ours | | PointNeXt |
|---|---|---|---|
| Input Type | NGLOD | iNGP | Point Cloud |
| $A_{CC}$ | 81.8 | 82.4 | 72.6 |
| Ret. Time (Naive) | 65.06 | 65.06 | 65.06 |
| Ret. Time (Hier. Samp.) Total | 36.19 | 35.46 | 35.78 |
| Ret. Time (Hier. Samp.) CD@128/4096 | 25.05 \| 11.14 | 25.05 \| 10.41 | 25.05 \| 10.73 |

Table 18: Category-Chamfer Retrieval Accuracy and Retrieval Time on ShapeNet10

As shown in Table 18, using Hierarchical Sampling significantly reduces the time (on average 1.8X) required for calculating the Chamfer Distance between different INRs. The speed-up for all methods is very similar as the point sampling and Chamfer Distance calculation time dominates the runtime. This leaves the difference in time for classification between the methods negligible. Using NGLOD as an example, the naive retrieval method involves point sampling and Chamfer Distance calculation (4096 points) for 49 INRs which costs 65.06 seconds, and an additional 0.04 seconds for classification. Using the hierarchical method, the distance point sampling and Chamfer Distance calculation are first done for 128 points (25.05 seconds + 0.04 seconds for classification), and around 17.1% of the INRs need to be further evaluated at 4096 points, resulting in a runtime of 11.14 seconds. We expect this speedup to scale further as more data is presented as the retrieval candidate. Despite the speed up, this process is still relatively slow compared to the category-only retrieval which typically only requires cosine similarity comparison. We leave potential methods that would allow fast and accurate Category-Chamfer retrieval as future work.

