# OpenReview forum: "INRet: A General Framework for Accurate Retrieval of INRs for Shapes"
_ICLR.cc/2024/Conference — Submitted to ICLR 2024_

### Official Review · Reviewer_rhWp · 2023-10-26

**Soundness:** 2 fair
**Presentation:** 2 fair
**Contribution:** 2 fair
**Rating:** 6
**Confidence:** 3

**Summary:**

This paper discusses a method to compare implicit neural representations, with the purpose of retrieving shapes similar to a query representation in a data base. The shapes in the database may be encoded using different category of representations (SDF, UDF, occupancy fields...) and neural representation (grid-based, full MLPs...). The problem is tackled by training encoders to generate one embedding from one representation (associated to one object). One such encoder is trained for each type of representation. The embeddings generated from the encoders are aligned by a specific loss including a) difference loss between embeddings generated from different representations of the same object and b) a similarity loss between the original and a reconstructed UDF representation. The latter is done by a single encoder, from the embedding, no matter the original representation. If no encoder exists for the type of representation of a shape, the paper proposes to train, using distillation, a representation that has such an encoder. Various experiments compare the proposed approach against other INR embedding methods and to converting to explicit representation to retrieve shapes.

**Strengths:**

originality:
- I am not aware of any similar work.

quality:
- Sensible baselines are used, in particular other INR-based approaches and using conversion and comparison in explicit form.
- The experiments show the strength of the approach.
- There is an ablation study.

clarity:
- The paper is well written and relatively clear.

Significance:
- The paper discusses an approach to cover representations not initially included and experiments for that case as well.

**Weaknesses:**

quality:
- In the baseline where retrieval is based on conversion to explicit representation, the quality of the reconstruction is not quantified, unless I missed it. Therefore it is not possible to know whether these baselines are worse than the proposed approach because the latter is superior or because the reconstruction is not good. Unless I missed something, an alternative may be to use the original meshes for retrieval rater than a reconstruction.

clarity:
- Figure 2 could benefit from the addition of symbols used in the text to denote the different elements, as done in Figure 1.

significance:
- I am not sure how likely a database containing INRs encoded using various types of INRs is.
- Based on my understanding of the approach, I think the encoding of the shapes in the database must be trained with the decoders used to create the INR embeddings (part b and c in Figure 2). Thus is not clear to me why handling multiple types of representation is necessary. If the database owner knows querying and retrieval is necessary (as part b and c have been trained), why not create an embedding using a single type of representation? Of course, encodings trained before the system is available can be handled by the distillation approach. In experiments 5.2 and 5.3, I was also not able to precisely understand whether the encodings were trained with or without the decoder

Typos/details:
- Figure 2: Emplicit
- For the same figure, I would like to suggest labeling the models that are mesh specific and global for the approach. Though not necessary, it may increase the clarity of the figure.

**Questions:**

I would be grateful to the authors for commenting on the weaknesses listed above, and answering the following question:
- As far as as know, features from different hierarchical levels of a grid-based INR are usually concatenated before being fed to the network. I was surprised to see that in this paper they are summed together (p5, top paragraph). Is there any advantage/downside to summing them?

---

> ### Author Response · Authors · 2023-11-23
>
> > __Reconstruction quality and its impact on retrieval performance__
>
> We provide details on the reconstruction quality achieved by various architectures in this work in Appendix C.1 Table 15. We also provide more visualization in Appendix C.2 Figure 4 to demonstrate the reconstruction quality.
>
> To understand how the retrieval accuracy is impacted by the reconstruction quality, we perform another experiment in Appendix C.1 (Table 16) where we train an iNGP INR with a small number of epochs (2) so that the iNGP INR has a lower reconstruction quality than an MLP-only INR. In summary, we found that despite having lower reconstruction quality than the MLP-only INR, the features in the iNGP hash grid still made retrieval accuracy from a low-reconstruction quality iNGP higher than the MLP-only INR. Therefore, supporting INRs with feature grid in INRet is essential for improving the retrieval accuracy irrespective of the reconstruction quality of the INRs. Please refer to Appendix C.1 Table 16 for more details.
>
> > __Meshes vs Reconstruction__
>
> We would like to note that for INRet, we use the mesh to train INRs and then perform retrieval using the features and weights in these trained INRs. Our goal in this work  is to analyze how retrieval can be done effectively if the shapes are implicitly represented with different kinds of INRs.
>
> If we were to support querying the INR database using meshes, there are two possibilities. Since INRet is designed to store INRs, to query INRs with the mesh, we can either convert the query mesh into an INR and use INRet, or reconstruct the INRs in the database and render into multi-view images and use view-based retrieval methods. As we have demonstrated in the experiments, converting the mesh to INR and use INRet is more accurate compared to multi-view-based methods (Section 5.2 Table 1).
>
> > __Figure 2 Labelling__
>
> We thank the reviewer for the suggestion and we have added additional labels/symbols in Figure 2.
>
> > __Database containing various types of INRs__
>
> INRs are becoming a very popular representation for shapes, graphics objects, scenes etc. For example, nerfstudio has recently published a tool to export trained NeRF scenes to unreal engine, which would make production use of these INR models much easier [1]. We also see companies starting to use INR to encode many forms of data, such as encoding fonts [2], materials [3] and various forms of 3D assets [4]. We expect to see more tools like this in the future, which means INRs will have an increasing impact in the real world. This would require a database to store these models for efficient retrieval, similar to relevant head searching in existing mesh-based databases [5].
>
> [1] Nerfstudio: Exporting to Unreal Engine [nerfstudio, Unreal Engine [https://docs.nerf.studio/extensions/unreal_engine.html]
>
> [2] Joint Implicit Neural Representation for High-fidelity and Compact Vector Fonts [ICCV2023, Adobe, https://openaccess.thecvf.com/content/ICCV2023/papers/Chen_Joint_Implicit_Neural_Representation_for_High-fidelity_and_Compact_Vector_Fonts_ICCV_2023_paper.pdf]
>
> [3] Real-Time Neural Materials using Block-Compressed Features [preprint, Ubisoft, https://hal.science/hal-04255874v1]
>
> [4] [Luma AI, https://lumalabs.ai/]
>
> [5] [Ubisoft, https://montreal.ubisoft.com/en/using-ml-and-complex-math-to-deconstruct-and-reconstruct-human-faces/]

---

> ### Author Response · Authors · 2023-11-23
>
> > __The need for encoder/decoder after the training process. The need for multiple representations__
>
> One notable feature of INRet is that the user models inserted into the database do not need to be trained along with the decoders that are used to help train the INR encoders. The user INRs can be trained from scratch by the user without having access to the encoder or decoder. To be more precise, the common decoder in Figure 2(c) is not needed for the embedding creation as new INR models are added to the database. The encoder in Figure 2(b) is frozen after the initial database setup, and is required when new INR models are added to the database. In summary, once the encoders are trained, given an INR pretrained by the user, the INR embedding can be created with a single forward pass through the trained encoders, which is much faster than converting the trained INR to other representations such as point cloud.
>
> We leave the option of creating an embedding using different INR representations because different INRs have different strengths. For example, as reviewer “U5A9” has pointed out, a multi-layer car might only be properly represented with a UDF INR. In the meanwhile, one may want to represent a watertight surface and ensure it is watertight so he/she may choose an SDF INR. We design INRet so that the users can feel free to use the best INR implicit function for the most suitable representation without worrying about the incompatibility between them for the retrieval process.
>
>
> > __Figure 2 mesh specific and global__
>
> INRet is not generally tied to mesh. We use mesh in this work only as a starting point to perform training of INRs. Figure 2(a) involves training of the INRs from mesh, however, specific INRs can also be trained from other 3D representations (UDF from point cloud, Occ from voxel grid). The encoders in Figure 2(b) are used for encoding new INRs put into the database, they are therefore global throughout the training and deployment phases. The decoder in Figure 2(c) is only used for the encoder training, and is not useful once the encoders have been successfully trained.
>
> > __Summation and concatenation of the features__
>
> We choose summation or concatenation of the features based on how the features were originally used in the grid-based INR. NGLOD used summation of the features and iNGP used concatenation of the features. One of the reasons for their design choice might be iNGP requires all the features to mitigate the negative effects of hash collision using the MLP, but NGLOD does not have the issue. INRet follows their design choice and uses the same method to combine features as input to the Conv3D encoder.
>
> We ran an additional experiment to test out what would happen if the features from NGLOD are concatenated (instead of summed), and the features from iNGP are summed (instead of concatenated). In summary, the retrieval accuracy significantly decreases, dropping 14.8% for NGLOD and 53.8% for iNGP. Please see further details in Appendix B.7.

---

> > ### Comment · Reviewer_rhWp · 2023-12-05
> >
> > Thank you for the update and clarification, especially the detailed comment on the motivation for the approach. I think the update makes the paper better and will upgrade my score to 6.

---

### Official Review · Reviewer_9nfq · 2023-10-30

**Soundness:** 4 excellent
**Presentation:** 4 excellent
**Contribution:** 4 excellent
**Rating:** 6
**Confidence:** 4

**Summary:**

This paper tackles the retrieval problem of Implicit Neural Representation (INR).

Existing methods perform retrieval with 3D data directly. As more and more INR data and converting to 3D data takes up some computation, this paper proposes a framework called INRet (INR Retrieve) to perform retrieval directly on INR space, supporting both MLP-only and MLP+Grid-based INR, including different implicit functions.

Specifically, the (query) input is INR weights and the grid-based features (e.g., InstantNGP hash grid). INR weights will be flatten and passed into an MLP encoder to obtain the embeddings, grid-based features will be passed into 3D convolution encoder to obtain the embeddings, the retrieval is then done by using the concatenated embeddings. During the training, the concatenated embeddings will also be concatenated with the coordinate inputs, and will be passed into an MLP implicit decoder. The training objective is to minimize the distance between the decoder output and the target function (e.g., SDF).

Additionally, conversion from conventional INR (without grid-based features) can be done through distillation (where the target is generated by the conventional INR).

**Strengths:**

originality:  compared to inr2vec [1], this paper proposes an additional module to handle the latest architecture of INR (i.e., octree-based and hash grid-based). This paper also describes how to handle different implicit functions (e.g., SDF, UDF, Occ) and to map the INRs into a common latent space, and proposes suitable regularization terms to minimize the domain gap between different implicit functions.

quality: the paper is technically sound.

clarity: the paper is well-organized and easy to understand.

significance: the paper is addressing a new topic -- retrieval on INRs. Different from conventional retrieval problems where embeddings are obtained directly from the input, which has some sense of interpretability, retrieval with INR is still in the black-box stage --- difficult to interpret the embeddings of INR. Hence, this paper is important to the community as this paper tackles how to retrieve directly in INR space, and future works may be inspired to solve the following limitations.

[1] https://arxiv.org/abs/2302.05438

**Weaknesses:**

This work designed a framework for retrieval with INRs and proposed regularization terms. However, the analysis of the proposed regularization terms only shows the performance comparison. It would be better if the paper could further analyze why these terms are so important.

A potential weakness of this work is that all data must be first encoded into the INR space. If given an unseen/test input that is a 3D model but wants to query similar INRs, how long to convert it into the INR? Since we can have many possible different INR weights that can reconstruct the 3D model, how to ensure that the embeddings of the INR weight are still meaningful? Possibly a tSNE of the learned embeddings can be provided.

The writing quality can be further improved, there are some typos and grammar errors (e.g., Sec 5.3 second paragraph, third line "respective datasets. also performs better".

**Questions:**

A potential weakness of this work is that all data must be first encoded into the INR space. If given an unseen/test input that is a 3D model but wants to query similar INRs, how long to convert it into the INR? Since we can have many possible different INR weights that can reconstruct the 3D model, how to ensure that the embeddings of the INR weight are still meaningful? Or is there any way to ensure this?

---

> ### Author Response · Authors · 2023-11-23
>
> > __The importance of regularization terms__
>
> We show tSNE plots highlighting the difference between INR embeddings when the common decoder is applied and is not applied. In summary, using the common decoder regularization leads to the embedding for the same category being much more tightly clustered. Without the common decoder, the embeddings for shapes belonging to the same category are much more spread out, this is likely a result of different decoders requiring the INR embedding for the same shape to be used for different purposes (decoding UDF, SDF, or Occ).  Please refer to Appendix D (Figure 5) for further details.
>
> > __tSNE plot; conversion and ensuring meaningful INR weight/embedding (also in Q1)__
>
> We provide tSNE plots including all of the UDF/SDF/Occ embeddings at the same time and separately in Appendix D (Figure 5 & 6), which helps to visualize the embedding space created by INRet.
>
> If the input is a 3D model (not INR), we apply the INR distillation technique detailed in Section 5.3 and Appendix A.3. It would take around 30 seconds to train a NGLOD or iNGP INR from the test 3D model to enable retrieval (Section 5.3 Table 2).
>
> To ensure the embeddings are meaningful, we require the INR’s MLP to be initialized in the same way as the INRs that were used for training the INR encoders. This is similar to inr2vec. We note that we do not have the same initialization requirement for feature grids as the feature grids are typically initialized with very small random values. This is further explained in Appendix A.1 INR Initialization.
>
> In the case that the INR MLP weights are initialized differently, we need to apply the INR distillation technique to train another INR with a specific MLP initialization. However, we note that using the same weight can be enabled via setting a constant random seed and does not affect the performance of the INR itself (such as the reconstruction quality).
>
> > __Writing Quality__
> We thank the reviewer for pointing out the mistakes and we have fixed them.

---

### Official Review · Reviewer_U5A9 · 2023-10-30

**Soundness:** 3 good
**Presentation:** 2 fair
**Contribution:** 2 fair
**Rating:** 5
**Confidence:** 4

**Summary:**

This paper presents INRet, a general framework for the retrieval of shape INRs. INRet uses the grid-based methods (e.g. NGLOD, instantNGP) as the backbone of INRs and is able to handle different INRs, such as SDF, UDF and Occ. INRet achieves the best results under shape retrieval task.

**Strengths:**

1. The task is quite interesting, which retrieves shapes in terms of INRs. This paper handles multiple INRs, such as SDF, UDF and Occ, which covers the commonly used implicit representations.

2. NGLOD and instantNGP are quite popular backbones, it is good to see that INRet supports these methods.

3. The evaluations seems good.

**Weaknesses:**

1. The matching among SDF, UDF and Occ may lead to errors. Both SDF and Occ can only represent watertight shapes, but researchers leverage UDF to represent open-surfaces and multi-layer structures. What if I input an INR in terms of a multi-layer car or a shirt with open surfaces? The matching from the SDF and Occ of watertight mesh of the car or the shirt is wrong for the UDF.

2. There is no visualizations for comparison or illustrating, which makes the paper lack of qualitative analysis. I also find it hard to get the advantages of INRet compared to other baselines without visualizations.

3. The authors mentioned that ``the prior work only supports MLP-based architecture, which is not commonly used today.'' I agree with the authors that grid-based approaches are more popular today, however, the MLP-based approaches are also commonly used today, and INRet do not support these methods. Furthermore, there are many other representations for INR today, can INRet also handle these types of INRs (e.g. triplane for TensoRF, point feature-based for Point-NeRF)?

**Questions:**

Please refer to the weaknesses above.

---

> ### Author Response · Authors · 2023-11-23
>
> > __Matching among SDF, UDF, Occ. Matching of multilayer car and open shirt__
>
> One of the main design goals of INRet is to support retrieval across INRs with different implicit functions, especially when the shape might be well represented with a specific kind of implicit function but not the others. While it might be difficult to find a “closed-surface” shirt counterpart to a open-surface shirt, we can easily demonstrate INRet’s robustness using multilayer car and single-layer watertight car.
>
> We demonstrate the robustness of INRet’s cross implicit function retrieval with a multilayer car and watertight car. In Appendix C.2 (Figure 4), we show that using a multilayer car as query (represented by a UDF INR) can successfully retrieve watertight cars (represented by SDF INR).
>
> > __Qualitative Visualizations__
>
> We have added more visualizations in Appendix C.2 to show qualitative comparison between our method and the baseline methods.
>
> > __INR with MLP-only, triplane and points__
>
> We note that INRet trivially supports encoding of MLP-only INRs by simply not encoding the feature grid component if it is lacking. We added more experiments in Appendix B.3 (Table 10) showcasing INRet’s performance when applied to MLP-only INRs. We have also included an evaluation for INRs with triplane feature grids which is supported by INRet (by simply encoding features from triplane grid with Conv3D encoder). For MLP-only INR, using INRet improves retrieval accuracy across different implicit functions drastically.  For triplane INR, using INRet leads to retrieval accuracy similar to NGLOD and iNGP INR. Please see Appendix B.3 (Table 10) for more details.
>
> We would like to note that since Point-NeRF uses points to perform novel view synthesis. The points alone cannot encode a continuous 3D field of SDF/UDF/Occ values and are thus not suitable for implicit shape representation.

---

### Official Review · Reviewer_upUQ · 2023-11-02

**Soundness:** 3 good
**Presentation:** 3 good
**Contribution:** 3 good
**Rating:** 6
**Confidence:** 4

**Summary:**

While the idea of Implicit Neural Representations (INR) for shapes has been proposed elsewhere, this paper focuses on the task of organization and retrieval of INRs. The authors propose INRet, a method for aggregating two primary INR architectures, the MLP encoder and the feature grid encoder. They create a new embedding, which they call the INR embedding, which is obtained by optimizing a signed distance loss over three different implicit functions (signed and unsigned distance functions and the occupancy field). The authors demonstrate superior retrieval accuracy over existing methods.

**Strengths:**

The method achieves a significantly higher retrieval accuracy ( > 10%) over the existing state of the art methods including inr2vec on ShapeNet and Pix3D datasets. The authors also show retrieval results on methods such a PointNeXt and View-GCN.

The method allows the multiple INR architectures as well as as different implicit function representations, and is thus general and flexible.

They design their loss cleverly to include two terms. One term is an L2 loss that minimizes the pairwise differences between the respective INR embeddings (for the same shape), even if they have different implicit functions. The other term is the mismatch between the decoder and the unsigned distance function. This strategy allows the authors to use a single common decoder for three different implicit functions as well as handle diverse INRs such as octree grids and hash grids.

The method define a common latent space for multiple implicit function representations of shapes. This is an interesting idea and needs to be developed further.

The authors also demonstrate retrieval performance on the converted explicit representations. This enables them to compare their results with PointNext and View-GCN.

The experimental results are sound and indeed demonstrate the effectiveness of the method.

**Weaknesses:**

The paper does not propose novel architectures or representations or even new metrics for similarity between the implicit representations.

Instead they make use of existing architectures and embeddings (separately for the MLP and the feature grid) as well as L2 losses for the pairwise dissimilarity

In equation (4), the two terms operate on different types of representations when they are compared. While one can trivially assume an Euclidean embedding (which the authors assume here), this may be problematic. Inside both the terms, the distance (d_u(x)) or the embeddings e_i, i \in {s, u, o} should be either appropriately weighted. This is especially important if the norms operate on an occupancy field as the distance functions and occupancy fields are structurally different quantities, the occupancy grid being more feature rich compared the the S/UDFs. This could perhaps be done naively by doing an internal conversion between the two before the norm is computed.

The above comment can also be understood in a different way. The INR embedding space is essentially a product space and thus a cost function that is used for matching two elements in that space, needs to be appropriately weighted. This type of an embedding is indeed interesting to explore and is one of the strengths of the paper (as listed above).


While the loss is designed to handle different types of input INRs, the decoder is restricted to outputting only a single type of INR. In this paper, it is likely the UDF. While this is fine for a  retrieval-only type application, which the authors demonstrate here, it makes the method slightly less general to be used for different applications (general shape embedding etc).

**Questions:**

In Equation (4), is the first loss term an L1 norm or the L2 norm? This is not clear and there will be different convergence properties of their encoder/decoder if it is an L1 norm.

In equation (4), since the first term compares the decoder outputs to d_u (an UDF), will the loss be biased towards UDFs? Can the authors comment? What if in the first term, the loss also incorporates pairwise differences between each INRs?

**Details Of Ethics Concerns:**

None.

---

> ### Author Response · Authors · 2023-11-23
>
> > __Novelty in proposing new architectures, representations, and metrics__
>
> Since the goal of our work is to demonstrate the feasibility of creating a common latent space for retrieval of INRs, we selected popular existing INR architectures and implicit functions without further modifications. This obviates the need for the use of new architectures or representations. Our hope is to keep the framework general such that it can easily support additional existing INR representations (such as triplane pointed out by reviewer U5A9) and future INR representations. Our analysis showed that embeddings to encode INRs can be generated using simple architectures such as Conv3D and MLP.
>
> In the updated version, we propose a new metric that addresses shortcomings of previously proposed metrics. We find that existing metrics evaluate the success of retrieval based on the category of the shape, while largely ignoring the differences between shapes within the category. Therefore, we propose a new metric: Category-Chamfer, where the retrieval is only successful if the retrieved shape is the shape that shares the same category while having the least Chamfer Distance with respect to the query shape. This was initially omitted from the paper since it not only applies to INR but to 3D shapes in general. We provide further details about this new metric as well as a way to accelerate retrieval of the shape required by this metric in Section 3.4 and Appendix E.
>
>
> > __Weighting and conversion between UDF/SDF/Occ embedding__
>
> In our original implementation, we used equal weighting for the explicit loss between UDF, SDF and Occ. We ran further experiments that used different weights between different explicit loss pairs. In summary, using different weights for the explicit loss has minimal impact on the final retrieval accuracy. Results and further explanations can be found in Appendix B.5.
>
> Regarding the potential conversion between UDF/SDF/Occ representations. We would like to note that the features INRet obtained from the feature grids are not actual UDF/SDF/Occ values. They are latent features used by the MLP of the INR for shape reconstruction, or used by the Conv3D encoder of INRet for INR embedding generation. Therefore, unlike the conversion between actual distance values (UDF/SDF/Occ) which is straightforward, there is no simple way to perform conversion between the latent vectors sampled from the feature grid. The difficulty of conversion in the latent space is one of the primary reasons we designed INRet to handle INRs with different implicit functions.
>
>
> > __Does occupancy contain more information than SDF/UDF__
>
> It is hard to quantify whether the occupancy grid is more feature-rich than the S/UDFs as they all ultimately serve to implicitly represent the surface of the shape. As shown by the quality of shape reconstruction in Appendix C.1 (Table 15) of the updated draft, Occ/SDF/UDF INRs have slightly different shape reconstruction quality (UDF>SDF>Occ) but they are in general at the same level.
>
> Theoretically, the SDF might contain the most information as it can be easily converted into UDF (by eliminating the sign) and Occ (by eliminating the magnitude). This may be why SDF-SDF retrieval (84.6 for iNGP) has higher accuracy than UDF-UDF retrieval (83.0 for iNGP) as seen in Section 5.4 (Table 3).
>
>
> > __Choice of loss for the common decoder; bias towards the loss of the common decoder (UDF); pairwise differences__
>
> We performed more experiments where we used a different type of loss for the common decoder (SDF and Occ). We provide experiment details and results in Appendix B.4 (Table 11). In summary, INRet is in general robust to the choice of common decoder loss. The retrieval accuracy is high regardless of the choice of the loss. This could enable applications such as shape generation if the trained decoder is used for this purpose. Please refer to Appendix B.4 (Table 11) for further details.

---

> ### Author Response · Authors · 2023-11-23
>
> > __Q1. L1 vs L2 norm for UDF common decoder in Equation (4)__
>
> The first loss is L1 norm. We chose L1 norm to be consistent with existing work that uses L1 norm for UDF loss (we used a UDF common decoder) [1]. We performed further experiments applying L2 norm to the UDF common decoder. In summary, the choice of L1 and L2 norm does not have a significant impact overall. Please refer to Appendix B.6 for further details.
>
>
> > __Q2. Potential bias towards the loss used by the common decoder (UDF); Pairwise differences__
>
> We performed further experiments to study the bias with respect to the choice of the common decoder loss. The choice of loss indeed makes the embedding slightly more biased towards the specific type of loss. For example, using the SDF common decoder, the SDF-SDF retrieval accuracy improved from 84.6% to 85.8%, while using the UDF common decoder resulted in the highest UDF-UDF retrieval accuracy. Please refer to Appendix B.4 (Table 11) for more details.
>
> We design a common decoder to output the same type of value regardless of the implicit function of the original INR. Therefore, the common decoder is not designed to take pairwise differences between the INRs. This is performed by the explicit L2 regularization.
>
>
> [1] Neural Unsigned Distance Fields for Implicit Function Learning [Neurips 2020, https://arxiv.org/abs/2010.13938]

---

### Author Response · Authors · 2023-11-23
**Overview of Changes**

We thank the reviewers for their insights and feedback. We have addressed all questions and concerns in our responses. We have updated the paper to address all feedback and included additional figures and experiments. We have highlighted the section/subsection head and figure/table captions of all new contents in orange.


Here is a summary of the content added.


Section 3.4, we introduce a new metric (Category-Chamfer) that considers both category information and similarities between shape instances within the same category.

Appendix B.3, experiments for other INR architectures (MLP-only, triplane)

Appendix B.4, experiments for different common decoder loss type

Appendix B.5, experiments for regularization terms weights

Appendix B.6, experiments for using L2 norm with UDF common decoder

Appendix B.7, experiments for using summation or concatenation of features from feature grids

Appendix C.1, experiments for reconstruction quality and impact of reconstruction quality on retrieval accuracy

Appendix C.2, visualization of retrieval

Appendix D, tSNE visualization of the embedding space

Appendix E, additional details and experiments for the new Category-Chamfer metric

---

### Meta-Review · Area_Chair_ZGFY · 2023-12-10

**Metareview:**

The submission introduces INRet, a framework for the retrieval of Implicit Neural Representations (INRs) of shapes, enabling direct retrieval of INR without explicit 3D data.
INRet covers different INR architectures, notably the MLP encoder and the grid-based methods, to deal with various INRs with different representations such as SDF, UDF, and Occupancy fields. The framework learns a shared INR embedding space across different INR types by contrastive learning to align them.
The proposed method shows a notable improvement in the retrieval performance.

The technical part of this work is largely based on the prior work, inr2vec, while extending the framework to support other popular INR structures, which is good. The authors propose a simple input design of the encoder to deal with grid-based INRs. This is a contribution of this work, but the training strategy is incremental to inr2vec.

Initially, the reviewers left concerns, including the following:

- Novelty (Reviewer upUQ)
- Artificial and limited setting (Reviewer rhWp)
- Writing and presentation (Reviewer 9nfq)
- Limited scope (Reviewer upUQ, 9nfq)

After the rebuttal, two reviewers checked the rebuttals, and one of the reviewers upgraded his/her score while another reviewer seemed lukewarm and kept the score (mentioned in a private comment). While the reviewers rated the score a bit positively, there are some parts to be rechecked in the reviewers' opinions.

**Justification For Why Not Higher Score:**

- Novelty: Indeed, this AC found that the framework of this work is largely based on the prior work, inr2vec.

- Writing and presentation: This AC found that all the reviewers seem to fail to understand the method in at least some parts and to be confused if not the authors' rebuttal. This AC read the revision and found that it was not satisfactory, in that the rebuttal answers were hard to find when reading the main paper of the current revision.

- Artificial and limited setting: The rebuttal does not directly answer the following question of Reviewer rhWp. The AC also agrees with Reviewer rhWp's concern on the practicality of the setting. It would have been practical if queries had heterogeneous modalities.
> "If the database owner knows querying and retrieval is necessary (as part b and c have been trained), why not create an embedding using a single type of representation?"


- Limited scope: It is ok to focus on retrieval only, but as Reviewer upUQ pointed out, it would be beneficial to demonstrate the impact of the work by showing its versatility through other applications (pointed out by Reviewer upUQ) or its generalization across cross-dataset generalization (pointed out by Reviewer 9nfq), like inr2vec. Since this work only focuses on a narrow scope of retrieval within each single and distinctive dataset, e.g., ShapeNet, it would have been good to demonstrate its practicality by setting more challenging and practical experiments, e.g., cross-dataset generalization for unseen categories.
Given these, compared to inr2vec, this work would have narrow interests by the community.

- While Reviewer 9nfq mentioned the INR retrieval topic is new, this comment is discounted as inr2vec already dealt with the same application.


Considering all the matters, this submission needs to be further improved for the ICLR publication. Since the work has a clear advantage of addressing grid-based INRs, the authors are encouraged to improve the work further and submit it to other conferences.

**Justification For Why Not Lower Score:**

.

---

### Decision · Program_Chairs · 2024-01-16

Reject